# Association between Antibiotic Use and Hospital-Onset *Clostridioides difficile* Infection in University Tertiary Hospital in Serbia, 2011–2021: An Ecological Analysis

**DOI:** 10.3390/antibiotics11091178

**Published:** 2022-08-31

**Authors:** Aneta Perić, Nemanja Rančić, Viktorija Dragojević-Simić, Bojana Milenković, Nenad Ljubenović, Bojan Rakonjac, Vesna Begović-Kuprešanin, Vesna Šuljagić

**Affiliations:** 1Department for Pharmacy, Military Medical Academy, 11000 Belgrade, Serbia; 2Medical Faculty, Military Medical Academy, University of Defence, 11000 Belgrade, Serbia; 3Centre for Clinical Pharmacology, Military Medical Academy, 11000 Belgrade, Serbia; 4Institute of Epidemiology, Military Medical Academy, 11000 Belgrade, Serbia; 5Institute of Medical Microbiology, Military Medical Academy, 11000 Belgrade, Serbia; 6Clinic for Infectious and Tropic Diseases, Military Medical Academy, 11000 Belgrade, Serbia; 7Department of Healthcare-Related Infection Control, Military Medical Academy, 11000 Belgrade, Serbia

**Keywords:** antibiotics, utilization, consumption, hospital, *Clostridioides difficile* infection, ecological study, developing country

## Abstract

This ecological study is the largest to date examining the association between rates of antibiotic use (AU) and hospital-onset (HO) *Clostridioides difficile* infection (CDI) in a tertiary university hospital in Serbia. There was no clear trend in the incidence of HO-CDI over time. Total utilization of antibacterials for systemic use increased from 38.57 DDD/100 bed-days (BD) in 2011 to 56.39 DDD/100 BD in 2021. The most commonly used antibiotics were third-generation cephalosporins, especially ceftriaxone, with maximum consumption in 2021 (19.14 DDD/100 BD). The share of the Access group in the total utilization of antibiotics ranged from 29.95% to 42.96% during the observed period. The utilization of the Reserve group of antibiotics indicated a statistically significant increasing trend (*p* = 0.034). A statistically significant difference in the consumption of medium-risk antibiotics from 2011 to 2021 was shown for penicillins and a combination of sulfamethoxazole and trimethoprim. The consumption of cefotaxime showed a statistically significant negative association with the rate of HO-CDI (r = −0.647; *p* = 0.031). Ampicillin and the combination of amoxicilline with clavulanic acid have shown a negative statistically significant correlation with the ID of HO-CDI (r = −0.773 and r = −0.821, respectively). Moreover, there was a statistically significant negative correlation between consumption of “medium-risk antibiotics” and the rate of HO-CDI (r = −0.677). The next challenging step for the hospital multidisciplinary team for antimicrobials is to modify the antibiotic list according to the Access, Watch, and Reserve classification, in such a way that at least 60% of the AU should be from the Access group, according to the World Health Organization recommendation.

## 1. Introduction

The use of antibiotics is one of the most important modifiable risk factors for the development of *Clostridioides difficile* infection (CDI), an infection that has become one of the most frequent healthcare-associated infections (HAI) worldwide [1,2]. Nearly every antibiotic has been associated with the development of CDI, including those used for the treatment of CDI—vancomycin and metronidazole [2]. Increasing risk is associated with a number of antibiotics, their dosage and duration of treatment. Antibiotic risks are heterogeneous. Certain antibiotic classes, such as cephalosporines, carbapenems, clindamycin, and fluoroquinolones, are more likely to cause CDI than other antibiotics [2,3,4,5]. Tetracyclines are a low-risk class of antibiotics [6].

A systematic review of existing literature describing the epidemiology and management of CDI in developing countries showed that the rate of community-associated (CA) and healthcare-associated (HA) CDI appears to be lower in developing countries than in developed ones, yet antibiotic exposure represents a major risk factor in both groups of countries [4,7]. A reduction in antibiotic use (AU), due to antibiotic stewardship programs, with an emphasis on the restriction of high-risk antibiotic use, resulted in a lower incidence of hospital-onset CDI (HO-CDI) [8]. According to Barlam et al., elements of antibiotic stewardship are the application of data on CDI incidence, as a measure of exposure to CDI, and AU, measured by the Defined Daily Dose (DDD) [9].

Another useful tool used in order to support monitoring of antibiotic prescribing is their classification into three groups: Access, Watch, and Reserve (AWaRe). The World Health Organization (WHO) proposes AWaRe classification in 2019 to balance AU with their restrictions [10]. A total of 180 antibiotics available around the world are classified into three groups and integrated into the Model List of Essential Medicines (EML) [11].

As we previously reported, the relationship between HO-CDI and AU is well established at the patient level [7,12], but institution-level risk factors for CDI are emerging. In this study, our aims were to evaluate AU and the incidence density (ID) of HO-CDI, as well as to explore the relationship between AU and the ID of HO-CDI within a tertiary university hospital in Serbia.

## 2. Results

A total of 869 episodes of HO-CDI occurred in the study period. The characteristics of these patients stratified by the years are shown in Table 1. The trend of the ID of HO-CDI is shown in Figure 1. The ID was the lowest during 2018 (2.0 per 10,000 patient-days) and the highest in 2019 (5.0 per 10,000 patient-days).

The statistical trends for total antibiotic utilization and shares of categories according to the WHO AWaRe classification over 11 years in our hospital are shown in Table 2. Total utilization of antibacterials for systemic use (J01 group) increased from 38.57 DDD/100 bed-days (BD) in 2011 to 54.68 in 2017, followed by a decrease in the next 3 years, and finally reaching 56.39 DDD/100 BD in 2021 (Table 2, Figure 2). Only the utilization of the Reserve group of antibiotics indicated a statistically significant increasing trend (*p* = 0.034) (Table 2, Figure 2).

The utilization of antibiotics in the Access group was less than 20 DDD/100 BD in the study period. The minimum was at the beginning of the study, in 2011 (12.12 DDD/100 BD), and the maximum was in 2013 (19.29 DDD/100 BD). The share of the Access group in the total utilization of antibiotics ranged from 29.95% in 2021 to 42.96% in 2013. Trends in consumption of individual antibiotics showed that only metronidazole had a consumption of 5 DDD/100 BD or more in the study period. The peak of utilization of metronidazole was in 2021 (10.31 DDD/100 BD). Amikacin had a utilization ≥2 DDD/100 BD in the analyzed period (Table 2). Following antibiotics, such as ampicillin, ampicillin + sulbactam, benzylpenicillin + procaine, benzylpenicillin, chloramphenicol, gentamicin, and trimethoprim + sulfamethoxazole showed a significant decrease in their use during the observed period. The utilization of antibiotics in the Watch group was less than 40 DDD/100 BD over an 11-year period, with its share from 56.90% in 2013 to 72% in 2019. Trends in consumption of individual antibiotics showed that ceftriaxone was the most commonly used in this group, ranging from 6.29 DDD/100 BD in 2012 to 19.14 DDD/100 BD in 2021. Cefotaxime and piperacillin + tazobactam showed a significant decrease in the use of antibiotics during the observed period. Vancomycin, meropenem, and cefuroxime had utilization ≥2 DDD/100 BD in the study period (Table 2). The minimum utilization of antibiotics in the Reserve group was 0.13% in 2011 and 2012. On the other hand, the highest utilization of antibiotics in the Reserve group was 2.73% in 2020. The mostly commonly used antibiotic in this group was colistimethate sodium (colistin) (around 1 DDD/100 BD from 2017 to 2021) (Table 2).

The correlation between consumption of individual antibiotics and ID of HO-CDI over an 11-year period is shown in Table 3.

The consumption of cefotaxime showed a statistically significant negative association with the ID of HO-CDI (r = −0.647). Ampicillin and the combination of amoxicilline with clavulanic acid have shown a negative statistically significant correlation with the ID of HO-CDI (r = −0.773 and r = −0.821, respectively) (Table 3).

The trend of medium-risk antibiotic consumption according to the risk of CDI from 2011 to 2021 was negative and statistically significant (Figure 3). A statistically significant difference in the consumption within this group of antibiotics during the observed period was shown for penicillins and the combination of sulfamethoxazole + trimethoprim (Table 4). Moreover, there was a statistically significant negative correlation between consumption of medium-risk antibiotics and the rate of CDI (r = −0.677) (Table 5, Figure 3).

The trend of vancomycin and metronidazole consumption was positive in the observed period (Table 2, Figure 4).

## 3. Discussion

This ecological study is the largest to date examining the association between rates of AU and HO-CDI in a tertiary university hospital in Serbia.

Balsells et al., estimated that the global overall ID for HO-CDI was 4.14 (95% CI = 3.10–5.53) per 10,000 patient-days, ranging from 1.67 (0.58–4.84) in Europe, 2.69 (1.32–5.49) in the Western Pacific, 6.36 (5.53–7.19) in North America, and a paucity of data from other regions of the world. Additionally, they emphasized that the cumulative incidence for the elderly was higher compared with the other age groups [13]. During the study period, the ID of our hospital corresponded to that registered in Europe (ranging from 2.0–5.0 per 10,000 patient days). The surveillance report based on data for 2016 retrieved from the European Surveillance System showed that the crude ID of HO-CDI was 2.4 cases/10,000 patient-days and the mean ID was the highest in tertiary care hospitals (5.8 cases/10,000 patient-days). Estonia, Lithuania, and Poland reported the highest ID of HO-CDI in Europe during 2016 [14].

A recently published systemic review of scientific papers in English from 2009 to 2019 (including territorial United Kingdom, France, Germany, Italy, Spain, Poland, US, Canada, Australia, Japan, and China) showed that there was no clear trend in the incidence of HO-CDI over time [15]. The results of our study were in accordance with that.

It is well known that older adults (≥65 years old) experience the greatest morbidity and mortality from CDI [16]. Loo et al., found that older age, use of antibiotics, and use of proton pump inhibitors (PPI) were significantly associated with HO-CDI [17]. Furthermore, Hensgens et al. demonstrated that the risk of developing CDI is 8 to 10-fold higher during antimicrobial therapy and 4 weeks afterwards, and 3-fold higher in the following two months [18]. Among our patients with HO-CDI, 65.3% were aged ≥65 years, 94.7% received antibiotics, and 32.2% received PPIs before the HO-CDI.

Monitoring of AU hospitals is crucial to identifying potential overuse, underuse, and inappropriate use. In order to improve the quality of AU in our hospital, a multidisciplinary healthcare team for antimicrobials (infectious disease specialist, clinical pharmacologist, hospital pharmacist, clinical microbiologist, and healthcare epidemiologist) was formed in 2008. The main aim of this team is to monitor the AU, especially broad-spectrum antibiotics. In a comparison of AU in our hospital from 2001–2010, when the average consumption was 56.6 ± 3.01 DDD/100 BD [19], we have noticed a 22% decrease in AU in the period from 2011 to 2021. This AU decline can probably be explained by the fact that the team for antimicrobials was formed, and constant meetings were held during which issues related to the consumption of antibiotics and the presence of HAI were analyzed and discussed.

Similar to the previous reports, cephalosporins were the most frequently prescribed in our hospital [20]. Our results showed that during the study, the most commonly used antibiotics were third-generation cephalosporins, especially ceftriaxone. Results from the study of AU in tertiary hospitals in Bangladesh showed that cephalosporins constituted around 50% of all antibiotics used, especially third-generation cephalosporins, with extensive use of ceftriaxone [21]. Use of this class of antibiotics is lower than 50% in our hospital. The maximum consumption of ceftriaxone was recorded in 2021 (19.14 DDD/100 BD), when the consumption of all antibiotics reached the value of 56.39 DDD/100 BD. According to the results from a worldwide point prevalence survey of hospital AU, ceftriaxone was the most commonly used antibiotic, ranging from 2.5% of prescriptions in Northern Europe to 24.8% in Eastern Europe [10]. Cephalosporines, except the first-generation, belong to the Watch group of antibiotics, as well as carbapenems and fluoroquinolones. In the last three years, antibiotics from the Watch group accounted for 58.03 to 72.01% of all prescribed antibiotics, while third-generation cephalosporins accounted for 15.56 to 35.7%. An increase in this group is concerning because the extensive use of third-generation cephalosporines is one of the factors contributing to the extensive spectrum of beta-lactamase producing microorganisms [22].

We observed a decrease in the use of cefazolin in our hospital in 2021 as a result of a decrease in elective surgical operations in favor of urgent surgical procedures, as well as difficulties in procuring antibiotics due to the COVID-19 crisis. A similar problem was reported by researchers from Portugal [23].

Namely, previous studies have suggested a strong association between cephalosporin use and CDI, and many national programs on CDI control have focused on reducing cephalosporin usage [24]. Despite reductions in cephalosporin use, however, rates of CDI have continued to rise. Generally, key risk factors for CDI were antibiotic choice, prescription of multiple antibiotics, and a long duration of treatment. Cephalosporins and clindamycin are most strongly associated with HA-CDI [5]. Possible explanations for the observed phenomena include the presence of comorbidities, polypharmacy, dose, and duration of antibiotic treatment, and the use of multiple antibiotics, sampling bias (commonly prescribed antibiotics will be more often reported as being associated with CDI), inappropriate controls and misclassification of *Clostridioides difficile*, clinical susceptibility bias, and the assumption that all antibiotics within a given class are equally associated with CDI risk [25]. Anyway, in our investigation, the consumption of cefotaxime showed a statistically significant negative association with the ID of HO-CDI (r = −0.647).

An overall 11-year increasing trend of carbapenem use, especially the use of meropenem, was documented in our study. The increase in meropenem consumption in 2020 and 2021 could be explained by the lack of infectious diseases specialists, the treatment of a large number of patients with advanced, severe forms of the underlying disease, and the reduction of monitoring of consumption of antibiotics in our hospital during the COVID-19 crisis.

Findings from the study of the pattern of AU in Serbia from 2010 to 2019 showed a statistically significant increase in quinolones use and an increase in the ratio of the broad-spectrum cephalosporins, macrolides, and penicillins compared to narrow-spectrum antibiotics. Those data mostly refer to AU in community healthcare, but prescribers regardless of where they work should be encouraged to include AWaRe classification in their routine practice [26].

Results from our study showed that the share of the Access group in the total utilization of antibiotics ranged from 29.95% to 42.96%. Pauwels et al. described patterns of worldwide hospital AU according to the AWaRe classification in the adult population [10]. They showed Access group use ranged from 28.4% in West and Central Asia to 57.7% in Oceania [10]. This study also registered differences in hospital consumption of antibiotics between regions in Europe. Access percentages in hospitals ranged from 30.2% in Eastern Europe to 55.2% in Northern Europe [10]. On the other hand, the overall use of the Reserve group in the world of adult inpatients was 2.0%, ranging from 0.03% in Africa to 4.7% in Latin America [27]. The use of antibiotics in the Reserve group in our hospital was from 0.13% to 2.73% during the study period. WHO introduced the target AU, stating that by the year 2023, at least 60% of the AU should be from the Access group [27]. The next challenging step for the hospital multidisciplinary team for antimicrobials is to modify the antibiotic list according to the Access, Watch, and Reserve classification, in such a way that at least 60% of AU should be from Access group, according to WHO recommendation.

Our data showed modest use of penicillins, although their use showed statistical significance. A possible explanation could be a shortage of some antibiotics in this group in Serbia. This is a common problem in low-and middle income countries, and one of the targets of WHO AWaRe classification is to find a way to resolve this problem [11].

Overuse of antibiotics in hospitals contributes to an increase in HO-CDI rates [1,5]. Kazakova et al. reported that reductions in total antibiotic use and use of third- and fourth-generation cephalosporins were each associated with decreased HO-CDI rates [28]. On the other hand, in England at the national level, a reduction in the use of fluoroquinolones has been shown to be associated with a decrease in the incidence of CDI caused by ribotype 027 [29], while in the US it was shown at the hospital level [30].

In our current analysis of AU, we showed decreased use of fluoroquinolones in 2016. The most frequently used was ciprofloxacin, followed by levofloxacin, while moxifloxacin has been used only for community acquired pneumonia and complicated skin and skin structure infections according to its summary of product characteristics in Serbia.

Similar to our findings, Yun et al. showed that ampicillin/sulbactam and cefotaxime positively correlated with the incidence of HO-CDI [8]. On the other hand, the association between HO-CDI and consumption of co-amoxiclav was reported by Vernaz et al. [31].

Preventing *Clostridioides difficile* transmission in hospitals is important and it is equally important that we achieve a better understanding of the factors influencing the risk of developing CDI, including antibiotic prescribing [24]. CDI characteristically occurs in elderly patients with comorbidities in whom the intestinal microbiota is disrupted due to antibiotic exposure. Reducing CDI risk by restricting the use of a small number of antibiotic classes may fail to reduce the overall incidence of CDI because those agents may be replaced by antibiotics with a similar risk of CDI. Thus, a balanced approach to antibiotic stewardship may be more beneficial (reducing unnecessary antibiotic use, reducing prolonged antibiotic duration, avoiding the use of multiple antibiotic classes, and promoting de-escalation of broad-spectrum therapy as soon as possible). Increasing the heterogeneity of antibiotic prescribing is associated with reduced selection pressure and the emergence of resistance [32,33].

Patients with CDI symptoms were treated with metronidazole and/or vancomycin during the study period. Previously, metronidazole was the first-line drug for non-severe CDI, while vancomycin was the drug of choice for severe CDI [34]. However, results of a study have shown the superiority of vancomycin compared with metronidazole. After that, Nelson et al. also concluded that metronidazole is inferior compared to vancomycin in the treatment of CDI [35]. Since 2017, according to the guidelines, fidaxomicin and vancomycin are the cornerstones of CDI treatment [36]. Fidaxomicin treatment in CDI patients results in lower recurrence rates compared to treatment with vancomycin and metronidazole, but it leads to higher acquisition costs [37]. However, fidaxomicin is not registered in Serbia and is not available for treatment of patients. According to recommendations from the year 2021, for the treatment of an initial episode of CDI, when fidaxomicin is not available, oral vancomycin in a dose of 125 mg, four times daily, for ten days is a suitable alternative. Oral metronidazole at a dose of 500 mg, three times daily, should be used only when vancomycin and fidaxomicin are not available [38]. Both drugs showed a trend of increasing consumption in our hospital due to the treatment of CDI as well as the use of these drugs in the treatment of other infections.

Our study has several limitations. First, this was a single-center study. In addition, because of ecological fallacy the study’s inferences may be correct but are only weakly supported by the aggregate data. Another major limitation was the lack of molecular typing data. The strength of our study is that differences in exposure to antibiotics at the hospital level may be bigger than at the patient level and so are more easily examined. The advantage of the study is also in the length of its duration. Indicators were continuously collected through organized surveillance for 11 years.

## 4. Materials and Methods

We conducted an ecological study at the Military Medical Academy (MMA), a teaching hospital of the University of Defense in Belgrade, Serbia, from 2011 to 2021. MMA is a 1200-bed tertiary healthcare center divided into 27 departments. Ethics approval for this study was granted by the Ethics Committee of the MMA (Project MF/MMA/02/20-22).

All adult patients (≥18 years) diagnosed with an initial episode of HO-CDI from January 1, 2011, to December 31, 2021, were included in the study. HO-CDI case were defined as any hospitalized patient with laboratory confirmation of a positive *Clostridioides difficile* toxin assay, associated with diarrhea (≥3 daily in a 24-h period with no other recognized cause) or visualization of pseudomembranes on sigmoidoscopy, colonoscopy, or histopathologic analysis on day three or later after admission to a MMA on day one [36,39,40]. We also included all patients readmitted to MMA. Readmission to MMA was defined as readmitted patients who did not have a CDI during their index admission to the hospital but had the onset of symptoms within four weeks of discharge from MMA [41]. First, CDI recurrence was defined as the return of symptoms associated with repeated positive tests within 15–56 days after the initial diagnosis was established [39]. Patients with CA-CDI and HA-CDI acquired in another hospital were excluded from the study.

In our tertiary healthcare center, positive testing for CDI is immediately reported to infectious diseases specialists and healthcare epidemiologists. A specialist in infectious diseases evaluates the best treatment options (metronidazole or vancomycin, or a combination of the two antibiotics). A healthcare epidemiologist evaluates the origin, risk factors, and outcomes of infection and recommends preventive measures to healthcare workers (HCW): (1) Maintain contact precautions for at least 48 h after diarrhea has resolved; adhere to hand hygiene practices with soap and water as well as daily patient bathing or showering with soap and water; (2) use dedicated patient-care equipment; and (3) perform daily cleaning of CDI patient rooms using a *Clostridioides difficile* sporicidal agent, etc.

Microbiological testing was performed at the Institute of Medical Microbiology at the MMA. During the study period, the diagnostic tests for *C. difficile* were the Automated EIA System for Toxins A/B (VIDAS CDAB) from 2011 to 2017 and the GeneXpert system test for detecting the presence of the tcdB gene (which encodes toxin B) of *C. difficile* from 2018 to 2021. A similar approach was used by authors in papers that also analyzed the impact of antibiotic consumption on HO-CDI in the US [42], as well as the epidemiology of CDI over longer periods of time in France [43].

Data on the use of antibiotics in inpatients were extracted from hospital software computer systems by hospital pharmacists in the Pharmacy Department. The number of grams or international units of antibiotics were converted into a number of DDD using the 2020 version of the ATC/DDD index. Data were expressed as DDD per 100 bed days (DDD/100 BD). DDD is the assumed average maintenance adult dose per day for the main indication of antibiotic, which refers to ATC code J01 (antibacterials for systemic use). The prescription of antimycotics for systemic use (J02) and antiviral drugs (J05) were excluded from the study. Antibiotics were labeled as “Access”, “Watch”, and “Reserve” using the 2019 WHO AWaRe Classification Database. The antibiotics in the Access group are considered as first- or second-line agents in the empiric treatment of a number of common infections and should be widely available. The Watch group includes antibiotics that have a risk of resistance, and they should be used as first- or second-options for a limited number of indications. They should be monitored and included in stewardship programs in hospitals. The Reserve group contains last-resort antibiotics that need to be intensively monitored and should be used only under certain specific conditions in order to preserve their effectiveness. The analysis focused on antibiotics and the risk of causing HO-CDI. We divided antibiotics into three groups based on their risk of causing CDI in patients, as defined in previous studies: high-risk antibiotics (cephalosporins, fluoroquinolones, clindamycin, and carbapenems), medium-risk antibiotics (penicillins, macrolides, aminoglycosides, sulfonamides, and trimetoprime) and low-risk antibiotics (tetracyclines) [3,4].

Based on standard statistical parameters (alpha error 0.05, minimum study power of 80%, and the two-tailed test), study power was calculated based on the Pearson correlation coefficient between total antibiotic consumption and CDI incidence according to the data literature (r = −0.112) [44], using the following correlation of a bivariate normal model (G*Power 3.1) of a minimum of 623 patients. To ensure sufficient power for the study, an eleven-year period was therefore taken for analysis.

Statistical analysis was performed with SPSS, version 26.0 (SPSS, Inc, Chicago, IL, USA). Results were expressed as absolute numbers with a proportion of the total number of each variable or mean with standard deviation (SD) for continuous variables. ID was defined as the number of HO-CDI per 10,000 patient-days. The correlation analysis between AU and ID HO-CDI was performed by Spearman correlation. A simple linear regression was carried out in order to assess the changes in trends of AU during the study period. The output of the software used (Excel data analysis module) includes the regression statistics of a linear regression analysis. In this statistical analysis, the results of each variable are presented as slope for trend, coefficient of determination, and the equation of the linear curve obtained on the basis of the fitted data. Figures were generated using Excel (Microsoft). Additionally, the Mann-Kendall statistical test was performed in the trend detection for antibiotic consumption and other analyzed variables.

## 5. Conclusions

In our study, we found that the incidence of HO-CDI did not increase, as well as the total consumption of antibiotics. At an ecologic level, in the tertiary university hospital, consumption of medium-risk antibiotics like penicillin, aminoglycosides, and macrolides had a significantly negative correlation with the rate of HO-CDI. The utilization of the Reserve group of antibiotics showed a statistically significant increasing trend. Therefore, more rational prescribing of antibiotics in our hospital is needed in the future.

## Figures and Tables

**Figure 1 antibiotics-11-01178-f001:**
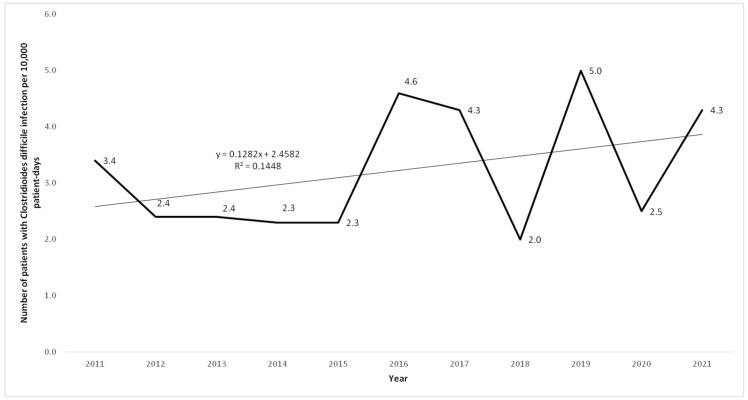
Incidence density (per 10,000 patient-days) of *Clostridioides difficile* in the tertiary university hospital in Serbia, 2011–2022.

**Figure 2 antibiotics-11-01178-f002:**
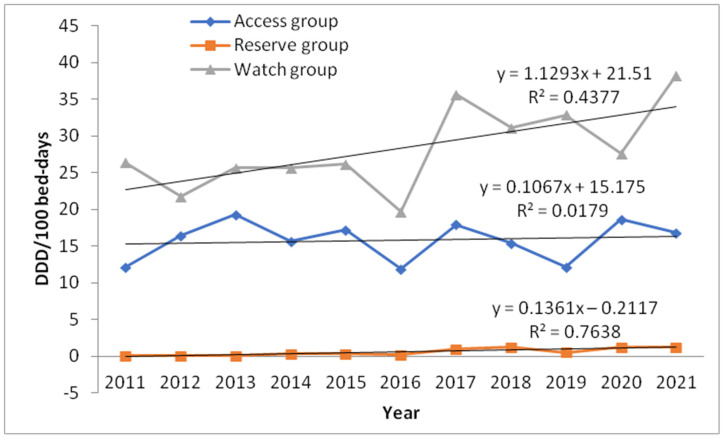
Trend of antibiotic consumption in defined daily doses (DDD) per 100 bed-days according to WHO Access, Watch, and Reserve Classification.

**Figure 3 antibiotics-11-01178-f003:**
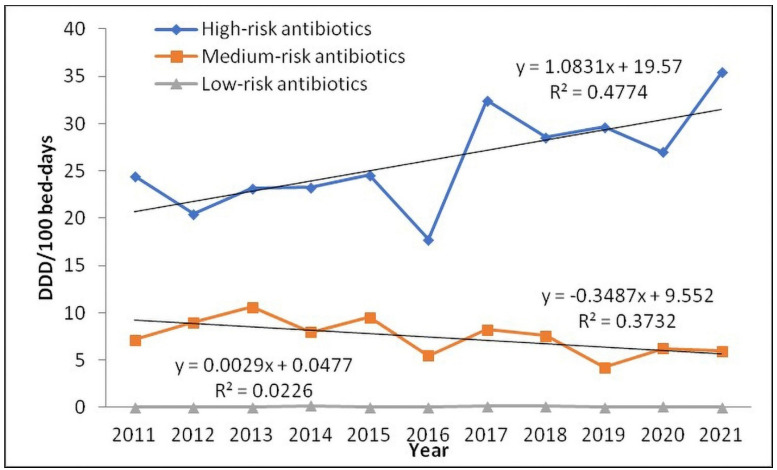
Trend in antibiotic consumption in defined daily doses (DDD) per 100 bed-days from 2011 to 2021 based on the risk of *Clostridium difficile* infection. High-risk antibiotics, Mann-Kendall test; *p* = 0.186; Medium-risk antibiotics, Mann-Kendall test; *p* = 0.010; Low-risk antibiotics, Mann-Kendall test; *p* = 0.873.

**Figure 4 antibiotics-11-01178-f004:**
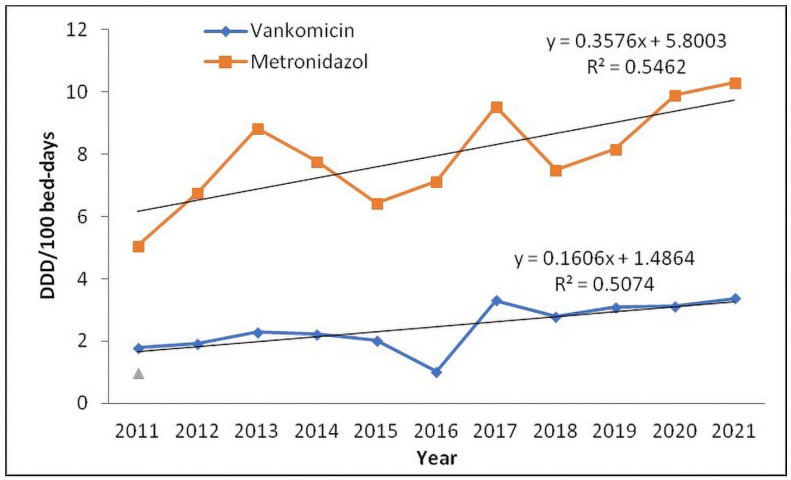
Trend of vancomycin and metronidazole consumption in defined daily doses (DDD) per 100 bed-days from 2011 to 2021.

**Table 1 antibiotics-11-01178-t001:** Characteristics of 869 patients with hospital-acquired *Clostridioides difficile* infection over 11-year period (2011–2021). CDI—*Clostridioides difficile* infection; AB-antibiotics; absolute number with percent or mean with standard deviation.

Characteristics	2011 *N* = 92 (%)	2012 *N* = 63 (%)	2013 *N* = 75 (%)	2014 *N* = 48 (%)	2015 *N* = 64 (%)	2016 *N* = 102 (%)	2017 *N* = 109 (%)	2018 *N* = 104 (%)	2019 *N* = 74 (%)	2020 *N* = 49 (%)	2021 *N* = 89 (%)	Total *N* = 869 (%)	*p*
**Male sex**	42 (45.7)	28 (44.4)	41 (54.7)	27 (56.2)	41 (64.1)	58 (56.9)	65 (59.6)	40 (38.5)	38 (51.4)	31 (63.3)	44 (49.4)	455 (52.4)	0.021
**Age**	65.20 ± 15.37	66.62 ± 13.74	65.05 ± 16.16	65.56 ± 17.69	72.81 ± 10.76	64.62 ± 14.97	68.56 ± 16.08	71.17 ± 14.09	68.47 ± 15.13	66.04 ± 19.64	68.75 ± 12.91	67.63 ± 15.22	0.006
**Age ≥ 65**	53 (57.6)	37 (58.7)	40 (53.3)	29 (60.4)	53 (82.8)	59 (57.8)	73 (67.0)	74 (71.2)	54 (73.0)	34 (69.4)	61 (68.5)	567 (65.2)	0.007
**Surgery**	50 (54.3)	28 (44.4)	33 (44.0)	23 (47.9)	29 (45.3)	43 (42.2)	62 (56.9)	38 (36.5)	22 (29.7)	19 (38.0)	33 (37.1)	380 (43.7)	0.012
**Intensive Care Unite**	20 (21.7)	12 (19.0)	7 (9.30)	12 (25.0)	21 (32.8)	24 (23.5)	30 (27.5)	22 (21.2)	22 (29.7)	18 (36.7)	21 (23.6)	209 (24.1)	0.033
**Nasogastric tube**	6 (6.5)	3 (4.8)	5 (6.7)	6 (12.5)	15 (23.4)	8 (7.8)	17 (15.6)	6 (5.8)	4 (5.4)	3 (6.1)	6 (6.7)	79 (9.1)	0.001
**Diabetes mellitus**	18 (19.6)	5 (7.9)	10 (13.3)	9 (18.8)	11 (17.2)	17 (16.7)	18 (16.5)	12 (11.5)	20 (27.0)	5 (10.2)	15 (16.9)	140 (16.1)	0.167
**Malignancy**	18 (19.6)	15 (23.8)	20 (26.7)	13 (27.1)	11 (17.2)	25 (24.5)	27 (24.8)	10 (9.6)	13 (17.6)	9 (18.4)	10 (11.2)	171 (19.7)	0.038
**Received AB prior to CDI**	87 (94.6)	58 (92.1)	71 (94.7)	45 (93.8)	63 (98.4)	98 (96.1)	103 (94.5)	94 (90.4)	70 (94.6)	47 (95.9)	86 (97.7)	822 (94.7)	0.556
**H2 antagonists**	65 (70.7)	25 (39.7)	17 (22.7)	12 (25.0)	15 (23.4)	39 (38.2)	47 (43.1)	26 (25.0)	17 (23.0)	3 (6.1)	1 (1.1)	267 (30.7)	<0.001
**Proton pump inhibitors**	36 (39.1)	15 (23.8)	18 (24.0)	12 (25.0)	17 (26.6)	36 (35.3)	28 (25.7)	34 (32.7)	26 (35.1)	26 (53.1)	32 (36.0)	280 (32.2)	0.017
**Chemotherapy**	4 (4.3)	12 (19.0)	5 (6.7)	4 (8.3)	1 (1.6)	12 (11.8)	3 (2.8)	4 (3.8)	9 (12.2)	5 (10.2)	2 (2.3)	61 (7.0)	<0.001
**Recurrence**	2 (2.2)	3 (4.8)	9 (12.0)	1 (2.1)	4 (6.2)	12 (11.8)	8 (7.3)	9 (8.7)	2 (2.7)	2 (4.1)	/ (0)	52 (6.0)	0.008

**Table 2 antibiotics-11-01178-t002:** Trends of antibiotic utilization over an 11-year period in the tertiary university hospital in Serbia according to WHO Access, Watch, and Reserve Classification. Consumption of particular antibiotics is expressed in defined daily doses (DDD) per 100 bed-days (BD).

	ATC Code		2011	2012	2013	2014	2015	2016	2017	2018	2019	2020	2021	Mann-Kendall Test
	J01	Total consumption of antibiotics	38.57	38.36	45.08	41.70	43.89	31.94	54.68	47.93	45.66	47.57	56.39	*p* = 0.138
ACCESS GROUP	J01GB06	Amikacin	2.95	3.66	4.15	3.19	4.51	1.34	4.32	4.10	2.06	3.44	3.56	*p* = 0.392
	J01CR02	Amoxicilline + Clavulanic acid	0.16	0.49	0.33	0.29	0.40	0.00	0.24	0.41	0.00	0.15	0.19	*p* = 0.083
	J01CA01	Ampicillin	0.34	1.05	2.29	0.89	1.64	0.25	0.47	0.64	0.21	0.37	0.17	*p* = 0.006
	J01CR01	Ampicillin + Sulbactam	0.07	0.00	0.15	0.00	0.06	0.00	0.06	0.00	0.00	0.00	0.00	*p* = 0.035
	J01CE09	Benzylpenicillin sodium + Procaine benzylpenicillin	1.06	0.70	0.66	0.44	0.34	0.00	0.06	0.04	0.02	0.02	0.01	*p* < 0.001
	J01CE01	Benzylpenicillin sodium (Penicillin G sodium)	0.00	0.00	0.00	0.00	0.00	0.00	0.00	0.02	0.02	0.00	0.00	*p* = 0.343
	J01DB04	Cefazolin	0.44	1.20	0.11	0.71	2.21	0.08	1.26	1.15	0.45	3.39	1.11	*p* = 1.000
	J01BA01	Chloramphenicol	0.01	0.00	0.00	0.01	0.03	0.01	0.00	0.00	0.00	0.00	0.00	*p* = 0.011
	J01FF01	Clindamycin	0.29	0.32	0.49	0.56	0.20	0.18	0.51	0.44	0.31	0.18	0.30	*p* = 0.139
	J01GB03	Gentamicin	1.35	1.49	1.47	1.10	1.06	2.81	1.01	0.76	0.64	0.80	0.80	*p* = 0.001
	J01XD01	Metronidazole	5.05	6.77	8.84	7.77	6.43	7.13	9.55	7.50	8.16	9.89	10.31	*p* = 0.102
	J01EE01	Trimethoprim + Sulfamethoxazole	0.40	0.80	0.80	0.67	0.38	0.18	0.48	0.39	0.37	0.45	0.43	*p* = 0.042
WATCH GROUP	J01FA10	Azithromycin	0.01	0.06	0.19	0.38	0.29	0.09	0.31	0.38	0.22	0.98	0.40	*p* = 0.073
	J01DC03	Cefuroxime	2.02	4.87	4.04	7.01	3.36	5.59	3.33	3.65	2.16	5.46	1.79	*p* = 0.102
	J01DD01	Cefotaxime	0.00	1.01	0.70	0.81	0.34	0.00	0.32	0.13	0.04	0.04	0.00	*p* = 0.018
	J01DD09	Ceftazidime	0.44	0.74	0.84	0.24	0.60	0.13	0.61	0.62	0.33	0.61	0.25	*p* = 0.139
	J01DD04	Ceftriaxone	15.68	6.29	10.77	8.01	10.60	6.59	15.21	11.81	15.94	6.74	19.14	*p* = 0.815
	J01DE01	Cefepime	0.08	0.00	0.00	0.07	0.66	0.01	0.49	0.46	0.35	0.69	0.38	*p* = 0.345
	J01MA02	Ciprofloxacin	2.33	1.25	1.12	0.40	0.21	0.19	2.48	0.55	1.70	2.37	3.09	*p* = 1.000
	J01FA01	Erythromycin	0.00	0.00	0.02	0.01	0.01	0.00	0.00	0.00	0.00	0.00	0.00	*p* = 0.079
	J01DH03	Ertapenem	0.58	0.56	0.80	1.17	2.22	1.63	1.72	1.52	0.80	1.25	1.21	*p* = 0.938
	J01DH51	Imipenem + Cilastatin	0.70	0.99	1.02	1.38	1.14	0.77	1.41	3.03	1.01	1.24	1.95	*p* = 0.243
	J01MA12	Levofloxacin	0.00	0.46	0.37	1.75	1.32	0.38	1.52	1.31	3.79	1.09	2.04	*p* = 0.274
	J01DH02	Meropenem	1.87	2.77	2.88	1.19	1.71	2.20	3.32	3.76	2.75	3.87	4.14	*p* = 0.139
	J01MA14	Moxifloxacin	0.00	0.00	0.00	0.00	0.00	0.00	0.30	0.11	0.00	0.07	0.03	*p* = 0.193
	J01CR05	Piperacillin + Tazobactam	0.83	0.76	0.56	1.01	0.85	0.79	1.31	0.83	0.69	0.06	0.41	*p* = 0.035
	J01XA02	Teicoplanin	0.07	0.17	0.06	0.07	0.94	0.28	0.08	0.23	0.00	0.00	0.00	*p* = 0.115
	J01XA01	Vancomycin	1.78	1.92	2.29	2.21	2.02	1.03	3.31	2.79	3.10	3.11	3.38	*p* = 0.102
**RESERVE GROUP**	J01XB01	Colistimethate sodium	0.02	0.03	0.08	0.09	0.17	0.15	0.74	0.97	0.46	1.05	1.10	*p* = 0.004
	J01XX08	Linezolid	0.00	0.00	0.01	0.11	0.19	0.10	0.11	0.19	0.07	0.17	0.16	*p* = 0.156
	J01AA12	Tigecycline	0.03	0.02	0.02	0.18	0.00	0.02	0.15	0.14	0.03	0.08	0.03	*p* = 0.697
		ACCESS GROUP	12.12	16.48	19.29	15.63	17.27	11.98	17.96	15.45	12.22	18.68	16.89	*p* = 0.697
		WATCH GROUP	26.39	21.83	25.67	25.70	26.25	19.70	35.73	31.18	32.88	27.59	38.21	*p* = 0.243
		RESERVE GROUP	0.05	0.05	0.11	0.38	0.36	0.26	1.00	1.30	0.56	1.30	1.29	*p* = 0.034

**Table 3 antibiotics-11-01178-t003:** Correlation between consumption of particular antibiotics and incidence density (ID) of *Clostridioides difficile* infection in the tertiary university hospital in Serbia, 2011–2021.

ATC Code	Antibiotics		Total Hospital ID of *Clostridioides difficile*
**J01MA02**	Ciprofloxacin	r	0.416
		*p*	0.203
**J01DD04**	Ceftriaxone	r	0.343
		*p*	0.301
**J01XD01**	Metronidazole	r	0.297
		*p*	0.374
**J01XA01**	Vancomycin	r	0.169
		*p*	0.619
**J01MA12**	Levofloxacin	r	0.165
		*p*	0.628
**J01DH02**	Meropenem	r	0.151
		*p*	0.658
**J01XB01**	Colistimethate Na	r	0.146
		*p*	0.667
**J01MA14**	Moxifloxacin	r	−0.027
		*p*	0.938
**J01GB03**	Gentamicin	r	−0.041
		*p*	0.904
**J01DE01**	Cefepime	r	−0.060
		*p*	0.862
**J01CR01**	Ampicillin + sulbactam	r	−0.064
		*p*	0.853
**J01CE01**	Benzylpenicillin sodium	r	−0.068
		*p*	0.843
**J01DH03**	Ertapenem	r	−0.096
		*p*	0.779
**J01AA12**	Tigecycline	r	−0.160
		*p*	0.638
**J01FA10**	Azithromycin	r	−0.206
		*p*	0.543
**J01BA01**	Chloramphenicol	r	−0.262
		*p*	0.436
**J01CR05**	Piperacillin + tazobactam	r	−0.265
		*p*	0.43
**J01FF01**	Clindamycin	r	−0.343
		*p*	0.301
**J01DB04**	Cefazolin	r	−0.348
		*p*	0.295
**J01XX08**	Linezolid	r	−0.356
		*p*	0.283
**J01DC03**	Cefuroxime	r	−0.357
		*p*	0.281
**J01EE01**	Trimethoprim + Sulfamethoxazole	r	−0.362
		*p*	0.273
**J01DH51**	Imipenem + cilastatin	r	−0.362
		*p*	0.275
**J01XA02**	Teicoplanin	r	−0.413
		*p*	0.207
**J01DD09**	Ceftazidime	r	−0.439
		*p*	0.176
**J01CE09**	Benzylpenicillin + Procaine benzylpenicillin	r	−0.513
		*p*	0.107
**J01FA01**	Erythromycin	r	−0.521
		*p*	0.100
**J01GB06**	Amikacin	r	−0.545
		*p*	0.083
**J01DD01**	Cefotaxime	r	−0.647
		*p*	0.031
**J01CA01**	Ampicillin	r	−0.773
		*p*	0.005
**J01CR02**	Amoxicilline + clavulanic acid	r	−0.821
		*p*	0.002

**Table 4 antibiotics-11-01178-t004:** The consumption of antibiotics according to the risk for *Clostridioides difficile* infection in defined daily doses (DDD) per 100 bed-days (BD) from 2011 to 2021.

		2011	2012	2013	2014	2015	2016	2017	2018	2019	2020	2021	Mann-Kendall Test
**High-risk antibiotics**	First-generation cephalosporins	0.44	1.20	0.11	0.71	2.21	0.08	1.26	1.15	0.45	3.39	1.11	*p* = 1.000
	Second-generation cephalosporins	2.02	4.87	4.04	7.01	3.36	5.59	3.33	3.65	2.16	5.46	1.79	*p* = 0.102
	Third-generation cephalosporins	16.11	8.05	12.32	9.06	11.54	6.73	16.15	12.56	16.32	7.40	19.39	*p* = 0.938
	Fourth-generation cephalosporins	0.08	0.00	0.00	0.07	0.66	0.01	0.49	0.46	0.35	0.69	0.38	*p* = 0.345
	Carbapenems	3.15	4.32	4.71	3.73	5.07	4.61	6.45	8.31	4.56	6.36	7.30	*p* = 0.139
	Fluoroquinolones	2.33	1.71	1.49	2.15	1.52	0.57	4.30	1.97	5.49	3.53	5.16	*p* = 0.586
	Clindamycin	0.29	0.32	0.49	0.56	0.20	0.18	0.51	0.44	0.31	0.18	0.30	*p* = 0.139
**Medium-risk antibiotics**	Penicillins	24.64	25.24	26.43	24.62	25.45	23.25	23.83	24.11	23.24	23.54	23.37	*p* = 0.002
	Aminoglycosides	4.30	5.16	5.62	4.28	5.56	4.15	5.33	4.86	2.70	4.23	4.36	*p* = 0.052
	Macrolides	0.01	0.06	0.20	0.38	0.30	0.09	0.31	0.38	0.22	0.98	0.40	*p* = 0.086
	Sulfamethoxazole and Trimethoprim	0.40	0.80	0.80	0.67	0.38	0.18	0.48	0.39	0.37	0.45	0.43	*p* = 0.042
**Low-risk antibiotics**	Tigecycline	0.03	0.02	0.02	0.18	0.00	0.02	0.15	0.14	0.03	0.08	0.03	*p* = 0.697

**Table 5 antibiotics-11-01178-t005:** Correlation between consumption of antibiotics classified according to the WHO Access, Watch, and Reserve classifications and the risk of *Clostridioides difficile* infection with the incidence density (ID) of *Clostridioides difficile* in the Serbian tertiary university hospital from 2011 to 2021.

	ID of *Clostridioides difficile*	
**Access group of antibiotics**	r	−0.275
	*p*	0.414
**Reserve group of antibiotics**	r	−0.037
	*p*	0.915
**Watch group of antibiotics**	r	0.261
	*p*	0.438
**High-risk antibiotics**	r	0.220
	*p*	0.516
**Medium-risk antibiotics**	r	−0.677
	*p*	0.022
**Low-risk antibiotics**	r	−0.160
	*p*	0.638
**TOTAL ANTIBIOTIC CONSUMPTION**	r	0.055
	*p*	0.873

## Data Availability

The data sets used and/or analyzed in the present study are available from the corresponding author on reasonable request.

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
