# Peer review of "Association between Antibiotic Use and Hospital-Onset Clostridioides difficile Infection in University Tertiary Hospital in Serbia, 2011–2021: An Ecological Analysis"

_antibiotics, 2022, doi:10.3390/antibiotics11091178_

Round 1
Reviewer 1 Report
The study is very interest and deals with an important issue.
The discussion is very long. I recommend to shorten it.
English editing should be performed
Author Response
Dear Reviewer 1,
We appreciate the careful review and constructive suggestions, which have been followed point by point. All changes made to the text are bolded in yellow. We do hope that our corrections notably improved the quality of the manuscript.
The study is very interest and deals with an important issue.
We thank Reviewer 1 for the comment.
The discussion is very long. I recommend to shorten it.
We thank Reviewer 1 for the suggestion. We removed the first paragraph of the discussion to shorten it. Accordingly, we excluded the first table from the results. Also, we excluded a few sentences from the discussion.
English editing should be performed
Many thanks to Reviewer 1 for the comment. We did English editing.
Reviewer 2 Report
The manuscript by Aneta Peric et al. describes an interesting study on the impact of C. difficile infection in university hospitals in Serbia over a ten-year period.
The manuscript deserves many modifications before possible acceptance for publication.
Global: prefer passive rather than active turns of phrase. "et al." and "e.g." in italics. A typo needs to be corrected.
Methods: how did the authors account for the bias associated with multiple testing? What multiple testing correction did they use?
Table 2: Were the authors able to stratify their results by year of diagnosis?
The figures are only captures of the excel image. The authors have to improve the quality.
Table 4: Could the authors consider stratifying their results according to the origin of the C.difficile infection?
Methods: How did the authors determine the number of subjects to include in the analysis?
Methods: How did the authors consider duplicates? Even in the absence of recurrence (what is the precise definition, by the way?), including the same patient twice is a major bias, and merits recalculation of all data considering this important change.
Methods: because these data may be biased by changes in diagnostic methods and treatment recommendations, authors should stratify their results by year of diagnosis.
Results: authors should not report results with a statistically insignificant association. Please review.
Discussion: According to ECCMID and American recommendations, the gold standard of treatment could not be Vancomycin and Metronidazole, but fidaxomicin.
Author Response
The manuscript by Aneta Peric et al. describes an interesting study on the impact of C. difficile infection in university hospitals in Serbia over a ten-year period.
We thank Reviewer 2 for the comment. Our manuscript describes the impact of antibiotic use on hospital onset of C. difficile infection in a university hospital in Serbia over eleven years.
The manuscript deserves many modifications before possible acceptance for publication.
We appreciate the careful review and constructive suggestions, which have been followed point by point. We do hope that our corrections notably improved the quality of the manuscript. All changes made to the text are bolded in yellow.
Global: prefer passive rather than active turns of phrase. "et al." and "e.g." in italics. A typo needs to be corrected.
We did English editing, correcting all typos and “et al.“ as well as “e.g“ are in italics in the whole manuscript. We believe that it has notably improved the quality of the manuscript.
Methods: how did the authors account for the bias associated with multiple testing? What multiple testing correction did they use?
Methods: How did the authors determine the number of subjects to include in the analysis?
Methods: How did the authors consider duplicates? Even in the absence of recurrence (what is the precise definition, by the way?), including the same patient twice is a major bias, and merits recalculation of all data considering this important change.
Many thanks to Reviewer 2 for the comments. We believe these questions refer to the same topic. We did not include duplicates, only the initial episode of HO CDI was included in the study as a case (“All adult patients (≥18 years) diagnosed with an initial episode of HO CDI from 1st January 2011 to 31st December 2021 were included in the study” - line number 326-332 in the revised manuscript). We have a separate database in which patients with HO CDI are entered, and we could very accurately observe duplicate patients. In the method, we emphasized how we defined recurrent cases (First CDI recurrence was defined as the return of symptoms associated with repeated positive test within 15–56 days after the initial diagnosis), and we also emphasized that these cases were excluded from further analysis of the incidence rate ( line 334-337 in the revised manuscript).
Table 2: Were the authors able to stratify their results by year of diagnosis?
We thank Reviewer 2 for this question. We did the stratification of our patient characteristics by year of diagnosis in Table 1.
The figures are only captures of the excel image. The authors have to improve the quality.
We thank Reviewer 2 for the comment. We improved the resolution of the figures. All figures are in JPEG format (300dpi)
Table 4: Could the authors consider stratifying their results according to the origin of the C.difficile infection?
Thank you for the question. A hospital epidemiologist assessed the origin of each CDI registered during the hospital treatment, and only HO CDI were included in this study.
Methods: How did the authors determine the number of subjects to include in the analysis?
Again, thank you for the question. “All adult patients (≥18 years) diagnosed with initial episode of HO CDI from 1st January 2011 to 31st December 2021 were included in the study. HO CDI case was defined as any hospitalized patient with laboratory confirmation of a positive toxin assay of Clostridioides difficile associated with diarrhoea (≥3 daily in a 24-hour period with no other recognized cause) or visualization of pseudomembranes on sigmoidoscopy, colonoscopy, or histopathologic analysis on day three or later, following admission to a MMA on day one” (line 326-332 in revised manuscript)
Methods: because these data may be biased by changes in diagnostic methods and treatment recommendations, authors should stratify their results by year of diagnosis.
Thank you, also, for this question. We have precisely defined which test was used during the specific time within the whole study period.
“During the study period, the diagnostic test for C. difficile was Automated EIA System for Toxins A/B (VIDAS CDAB) from 2011 to 2017 and GeneXpert system test for detecting the presence of the tcdB gene (which encodes toxin B) of Clostridium difficile from 2018 to 2021.” (line 348-353 in the revised manuscript)
A similar approach was used by authors in papers that also analyze the impact of antibiotic consumption on HO CDI (Webb BJ, Subramanian A, Lopansri B, Goodman B, Jones PB, Ferraro J, Stenehjem E, Brown SM. Antibiotic exposure and risk for hospital-associated Clostridioides difficile infection. Antimicrob Agents Chemother 2020 64:e02169-19. https://doi.org/10.1128/AAC.02169-19.), as well as the epidemiology of CDI over longer periods of time (Colomb-Cotinat M, Assouvie L, Durand J, Daniau C, Leon L, Maugat S, Soing-Altrach S, Gateau C, Couturier J, Arnaud I, Astagneau P, Berger-Carbonne A, Barbut F. Epidemiology of Clostridioides difficile infections, France, 2010 to 2017. Euro Surveill. 2019 Aug; 24(35):1800638. doi: 10.2807/1560-7917.ES.2019.24.35.1800638. PMID: 31481147; PMCID: PMC6724465.)
Results: authors should not report results with a statistically insignificant association. Please review.
Thank you for this comment. We changed our abstract according to it.
"This ecological study is the largest to date examining the association between rates of antibiotic use (AU) and hospital-onset (HO) Clostridioides difficile infection (CDI) in the tertiary university hospital in Serbia. The overall 11-year increasing trend of HO CDIs did not prove to be statistically significant (p=0.087). Total utilization of antibacterials for systemic use increased from 38.57 DDD/100 bed-days (BD) in 2011 to 56.39 DDD/100 BD in 2021. The most used antibiotics were third-generation cephalosporins, especially ceftriaxone with maximum consumption in 2021 (19.14 DDD/100 BD). The share of the Access group in the total utilization of antibiotics ranged from 29.95% to 42.96% during the observed period. The utilization of the reserve group of antibiotics indicated a statistically significant increasing trend (p=0.034). A statistically significant difference in the consumption of medium-risk antibiotics from the year 2011 to 2021 was shown for penicillins and a combination of sulfamethoxazole and trimethoprim. The consumption of cefotaxime showed a statistically significant negative association with the rate of HO-CDI (r= - 0.647; p=0.031). Ampicillin and a combination of amoxicilline with clavulanic acid have shown a negative statistically significant correlation with the ID of HO-CDI (r= - 0.773 and r= - 0.821, respectively). Moreover, there was a statistically significant negative correlation between the consumption of "medium-risk antibiotics" and the rate of HO-CDI (r = - 0.677). The next challenging step for the hospital multidisciplinary team for antimicrobials is to modify the antibiotic list according to the Access, Watch, and Reserve classification, in such a way that at least 60% of AU should be from the Access group, according to the World Health Organization recommendation."
Discussion: According to ECCMID and American recommendations, the gold standard of treatment could not be Vancomycin and Metronidazole, but fidaxomicin.
Thank you, also, for this comment. We try to explain the possibility of treatment in our country.
“Patients with CDI symptoms were treated with metronidazole and/or vancomycin during the study period. Previously, metronidazole was the first-line drug in non-severe CDI, while vancomycin was the drug of choice for severe CDI [34]. However, the results of the study have shown the superiority of vancomycin compared with metronidazole. After that, Nelson et al. also concluded that metronidazole is inferior compared to vancomycin in the treatment of CDI [35]. Since 2017, according to the guidelines, fidaxomicin and vancomycin are the cornerstones of CDI treatment [36]. Fidaxomicin treatment in CDI patients results in lower recurrence rates in according to treatment with vancomycin and metronidazole, but it leads to higher acquisition costs [37]. However, fidaxomicin is not registered in Serbia and is not available for the treatment of patients. According to recommendations from the year 2021, for the treatment of an initial episode of CDI, when fidaxomicin is not available, oral vancomycin in a dose of 125mg, four times daily, for ten days is a suitable alternative. Oral metronidazole in a dose of 500mg, three times daily, should be used only when vancomycin and fidaxomicin are not available [38]”
Reviewer 3 Report
This is an ecological study correlating the use of antibiotics with the incidence of infection by C. difficile.
The summary is satisfactory and contains the data of the work, but the conclusion about the Acces, Watch and Reserve system leaves doubts for readers who do not know. Maybe it could be improved in the summary.
In the methodology explain what are patients treated. Are patients who received antimicrobials? Likewise what are deceased patients? Patients who would progress to death?
Table 2 needs typo correction in words "unit" and "absolute"
There are two possible denominators for calculating the DDD: 1,000 patient-days and 100 bed-days. Please make it clear that in the comparisons you used the same denominator.
And in the results, the authors found a statistically significant difference in the reserve antibiotics group (p=0.03) but in table 5 there is neither colistin nor linezolid. Please explain the reason
Author Response
We appreciate the careful review and constructive suggestions, which have been followed point by point. All changes made to the text are bolded in yellow. We do hope that our corrections notably improved the quality of the manuscript
The summary is satisfactory and contains the data of the work, but the conclusion about the Acces, Watch, and Reserve system leaves doubts for readers who do not know. Maybe it could be improved in the summary.
Thank you for the comment. We changed our summery.
“The next challenging step of the hospital multidisciplinary team for antimicrobials is to modify the antibiotic list according to the Access, Watch, and Reserve classification, in such a way that at least 60% of AU should be from Access group, according to WHO recommendation.”
In the methodology explain what are patients treated.
Thank you for the comment. “All adult patients (≥18 years) diagnosed with an initial episode of HO CDI from 1st January 2011 to 31st December 2021 were included in the study. HO CDI case was defined as any hospitalized patient with laboratory confirmation of a positive toxin assay of Clostridioides difficile associated with diarrhoea (≥3 daily in a 24-hour period with no other recognized cause) or visualization of pseudomembranes on sigmoidoscopy, colonoscopy, or histopathologic analysis on day three or later, following admission to a MMA on day one” (line 326-332 in the revised manuscript)
Are patients who received antimicrobials?
Thank you for asking. We did the stratification of our patients characteristics by year in Table 1. Now you can see how many patients received antibiotics before the onset of HO CDI by year.
Likewise what are deceased patients? Patients who would progress to death?
Again, thank you for the question. We removed the first paragraph of the discussion to shorten it by the suggestion of Reviewer 1. Accordingly, we excluded the first table from the results and there are no deceased patients in the manuscript, now.
Table 2 needs typo correction in words "unit" and "absolute"
Many thanks to Reviewer 3 for the comments. We did English editing and corrected “unite” and “absolute”
There are two possible denominators for calculating the DDD: 1,000 patient-days and 100 bed-days. Please make it clear that in the comparisons you used the same denominator.
Many thanks to the Reviewer for this comment. The use of antibiotics in inpatients was expressed as DDD per 100 bed days (DDD/100 BD). On the other hand, incidence density (ID) was defined as the number of hospital-onset Clostridioides difficile infections (HO-CDI) per 10000 patient-days.
And in the results, the authors found a statistically significant difference in the reserve antibiotics group (p=0.03) but in table 5 there is neither colistin nor linezolid. Please explain the reason
Many thanks to Reviewer for this comment. Based on the work "Pauwels, I.; Versporten, A.; Drapier, N.; Vlieghe, E.; Goossens, H.; Global-PPS network Hospital Antibiotic Prescribing Patterns in Adult Patients According to the WHO Access, Watch and Reserve Classification (AWaRe): Results from a Worldwide Point Prevalence Survey in 69 Countries. J Antimicrob Chemother 2021, 76, 1614–1624, doi:10.1093/jac/dkab050.", an accurate division of antibiotics into three groups based on the risk of CDI was recommended. Colistin and linezolid are not on this list, along with other reserve antibiotics, and therefore these antibiotics were not listed in Table 5.
Round 2
Reviewer 2 Report
Dear Authors,
You have addressed most of my preivous comments but some remains (please see thereafter).
Methods: how did the authors account for the bias associated with multiple testing? What multiple testing correction did they use?
--> My comment was not completely understood. I refer to the risk of alpha risk inflation, due to multiple statistical testing on the same cohort. Multiple testing corection must be considered (as Bonferonni's for example).
Methods: How did the authors determine the number of subjects to include in the analysis?
--> Have the authors determined an a priori number of patients to include? If not, could they determine the statistical power of their study?
Results: authors should not report results with a statistically insignificant association. Please review.
--> This comment is also of interest for the main manuscript (and not the abstract alone)
Author Response
Methods: how did the authors account for the bias associated with multiple testing? What multiple testing correction did they use?
--> My comment was not completely understood. I refer to the risk of alpha risk inflation, due to multiple statistical testing on the same cohort. Multiple testing corection must be considered (as Bonferonni's for example).
Response: A correction made to p values when several dependent (or) independent statistical tests are being performed simultaneously on a single data set is known as Bonferroni correction. However, given that only two variables, CDI incidence and antibiotic consumption, are analyzed simultaneously, no correction is needed. Also, the Mann-Kendall test only works with one set of data across multiple time points, so no p-value correction is necessary here.
Methods: How did the authors determine the number of subjects to include in the analysis?
--> Have the authors determined an a priori number of patients to include? If not, could they determine the statistical power of their study?
Response: Determining the sample size and the power of the study was done before defining the patient follow-up period, i.e. taking the minimum number of patients in order to obtain representative results. Thank you for this comment. We have included this assessment in the methodology section.
Based on standard statistical parameters (alpha error 0.05, minimum study power of 80%, two-tails test) study power was calculated based on the Pearson correlation coefficient between total antibiotic consumption and CDI incidence according to the literature (r=-0.112) (Jachowicz E, Różańska A, Pobiega M, Topolski M, Wójkowska-Mach J. Consumption of Antibiotics and Epidemiology of Clostridioides difficile in the European Union in 2016-Opportunity for Practical Application of Aggregate ECDC Data. Antibiotics (Basel). 2020;9(3):127. doi: 10.3390/antibiotics9030127.) using Correlation: Bivariate normal model (G*Power 3.1) of a minimum of 623 patients. To ensure sufficient power of the study, an eleven-year period was therefore taken for analysis.
Results: authors should not report results with a statistically insignificant association. Please review.
--> This comment is also of interest for the main manuscript (and not the abstract alone)
Response: Thank you for this comment. Everything that was not significant was deleted from the description of the results as well as from the discussion.
Round 3
Reviewer 2 Report
The manuscript has been greatly improved by the authors following my previous comments.